# Factors Predictive of Mortality among Geriatric Patients Sustaining Low-Energy Blunt Trauma

**DOI:** 10.3390/healthcare10112214

**Published:** 2022-11-04

**Authors:** Nikhil Patel, Timothy N. Le, Seleshi Demissie, Shreya Pandya, Thomas Kania, Michael Copty, Sara Alothman, Amy Rost, Chris Governo, Frank DiRoma, Galina Glinik, Krassimir Atanassov, Boris Khodorkovsky, Anita Szerszen, Asaf Gave, Duraid Younan

**Affiliations:** 1The Department of Surgery, Division of Acute Care Surgery, Staten Island University Hospital, Staten Island, New York, NY 10305, USA; 2The Department of Surgery, SUNY Downstate Hospital, Brookyln, New York, NY 11203, USA; 3Biostatistics Unit, Feinstein Institutes for Medical Research, Staten Island University Hospital, Staten Island, New York, NY 10305, USA; 4Department of Emergency Medicine, Staten Island University Hospital, Staten Island, New York, NY 10305, USA; 5Department of Medicine, Division of Geriatric Medicine, Staten Island, New York, NY 10305, USA

**Keywords:** blunt, trauma, geriatric

## Abstract

Background: In geriatric trauma patients, higher mortality rate is observed compared to younger patients. A significant portion of trauma sustained by this age group comes from low-energy mechanisms (fall from standing or sitting). We sought to investigate the outcome of these patients and identify factors associated with mortality. Methods: A retrospective review of 1285 geriatric trauma patients who came to our level 1 trauma center for trauma activation (hospital alert to mobilize surgical trauma service, emergency department trauma team, nursing, and ancillary staff for highest level of critical care) after sustaining low-energy blunt trauma over a 1-year period. IRB approval was obtained, data collected included demographics, vital signs, laboratory data, injuries sustained, length of stay and outcomes. Patients were divided into three age categories: 65–74, 75–84 and >85. Comorbidities collected included a history of chronic renal failure, COPD, Hypertension and Myocardial Infarction. Results: 1285 geriatric patients (age > 65 years) presented to our level 1 trauma center for trauma activation with a low-energy blunt trauma during the study period; 34.8% of the patients were men, 20.5% had at least one comorbidity, and 89.6% were white. Median LOS was 5 days; 37 (2.9%) patients died. Age of 85 and over (OR 3.44 with 95% CI 1.01–11.7 and 2.85 with 95% CI 1.0–6.76, when compared to 65–74 and 75–84, respectively), injury severity score (ISS) (OR 1.08, 95% CI 1.02 to 1.15) and the presence of more than one comorbidity (OR 2.68, 95% CI 1.26 to 5.68) were independently predictive of death on multi-variable logistic regression analysis. Conclusion: Age more than 85 years, higher injury severity score and the presence of more than one comorbidity are independent predictors of mortality among geriatric patients presenting with low-energy blunt trauma.

## 1. Introduction

Traumatic injuries are the 5th leading cause of death in the older population and the mortality risk increases steadily with increasing age despite a decrease in injury severity [1,2]. Factors which impact mortality in the older trauma population are not well understood, even as trauma in older adults continues to emerge as a global health concern. Geriatric trauma is also frequently associated with complications despite a generally lower energy mechanism; these complications increase the odds of death [3].

Previous work showed that with every 1-year increase in age over 65, the odds of death after trauma increase by as much as 6.8%. Low-energy fall-related head injury can be associated with significant functional decline and increased resource utilization, with those older than 80 years having 1.6 times greater chance of dying than patients aged 65–80 years. The eastern surgical association (east) practice management guidelines proposed a lower threshold for trauma activation (hospital alert to mobilize surgical trauma service, emergency department trauma team, nursing, and ancillary staff for highest level of critical care) for injured patients aged 65 years or older, aggressive triage, correction of coagulopathy, and limitation of care when clinical evidence predicts an overwhelming likelihood of poor long-term prognosis [4]. 

Moreover, outcomes from these injuries display stark differences once age is compared. Besides head trauma, rib and pelvic fractures are the most common manifestations of low-energy blunt trauma (falls from standing, sitting), both significantly increasing the mortality of this population. One of the differences is seen in head trauma as the geriatric population has twice the mortality of younger patients; in addition, the number of rib fractures is directly linked to need for higher level care units and mortality, whereas hip fractures can often go unseen on plain films. Thus, elderly trauma care can be very challenging and should be treated uniquely [5,6]. Since low-energy trauma accounts for around 75% of all elderly trauma and triage is often difficult because of a multitude of factors such as deafness, dementia, and circumstances, as these falls are often unwitnessed, it is important that the factors predictive of worse outcomes are identified [5,6]. We sought to identify the factors predictive of death among geriatric patients sustaining low energy blunt trauma in a large level 1 trauma center.

## 2. Materials and Methods

### 2.1. Patient Selection and Variable Definition

After obtaining the Institutional Review Board (IRB) approval, a retrospective analysis of geriatric trauma patients (65 years and older) who arrived to our trauma center (Staten Island University Hospital Center in Staten Island, New York, NY, USA) for trauma activation (hospital alert to mobilize surgical trauma service, emergency department trauma team, nursing, and ancillary staff for highest level of critical care) between January 2019 and October 2019 was conducted using the trauma database. All patients received computed tomography scans (CT Scans) of the head, cervical spine, chest, abdomen and pelvis.

### 2.2. Inclusion and Exclusion Criteria

All geriatric trauma patients who arrived for trauma activation (hospital alert to mobilize surgical trauma service, emergency department trauma team, nursing, and ancillary staff for highest level of critical care) over the study period were considered for inclusion. Data collected from the trauma data base included demographics (age, gender, and race), mechanism of injury (blunt, low-energy), injuries sustained, injury severity (injury severity score “ISS”), transfused blood products including 24 h packed cells, transfused total admission packed red blood cells, fresh frozen plasma and platelets, and presenting vital signs. We also collected data on pre-existing comorbidities (chronic renal failure “CRF”, diabetes mellitus “DM”, hypertension “HTN” and chronic obstructive pulmonary disease “COPD”), hospital length of stay (LOS), disposition and survival. The patients were divided into three age groups: 65–74, 75–84 and ≥85.

### 2.3. Statistical Analysis

This is a retrospective cohort study. Categorical data were summarized by the number and percentage of patients falling within each category. Continuous variables were summarized by descriptive statistics including mean and standard deviation or median and interquartile range. The primary comparison group is age group (65–74, 75–84, 85+). Age is an independent predictor of death in trauma patients; we wanted to assess the impact age has on outcomes of geriatric (65 years of age and older) patients by categorizing them into three age groups for ease of interpretation and assessment into 65–74, 75–84 and 85+ age groups. Having the patients belong to three 10-year categories enables ease of interpretation and management. The primary outcome variable is hospital mortality. Bivariate analyses were performed using the χ^2^-test, ANOVA and Wilcoxon Rank-sum test, as appropriate. The independent effects of age group, Injury Severity Score (ISS) and previous medical history (whether patient has two or more comorbidities) were evaluated using a multivariable logistic regression analysis. All statistical tests were two-sided. *p*-values < 0.05 were considered statistically significant. All statistical analyses were performed using SAS software (Statistical Analysis Systems Inc., Cary, NC, USA). Confidence intervals (CIs) were two-sided, unless otherwise stated. This was done to evaluate if age and the presence of more than one comorbidity were independently predictive of death (at hospital discharge) on multi-variable logistic regression analysis.

## 3. Results

There were 1285 geriatric patients (age > 65 years) that presented to our level 1 trauma center for trauma activation (hospital alert to mobilize surgical trauma service, emergency department trauma team, nursing, and ancillary staff for highest level of critical care) with a low-energy blunt trauma (fall from sitting or standing) during the study period; 34.8% of the patients were men, 89.6% were white and 20.5% had at least one comorbidity. The median injury severity score (ISS) was 5 and the median hospital length of stay (LOS) was 5, too. Injury and clinical characteristics are demonstrated in Table 1. Thirty-seven (2.9%) patients died with mortality determined to be due to the trauma itself.

When the patients were divided into three age groups, significant differences were noted between the groups in sex (*p* = 0.01), race (*p* < 0.01), injury to extremities (*p* < 0.01), smoking (*p* < 0.001) and disposition (*p* < 0.0001) (Table 2).

While patients in the 65–75 age group had a higher rate of COPD (*p* = 0.01) compared to the two other age groups, there was no significant difference in the rates of CRF, HTN or MI.

Patients who died had a significantly higher incidence of chronic renal failure (CRF) *p* = 0.02 compared to those who did not; there was no significant difference in the rates of COPD, MI or HTN between the two groups.

Age of 85 or more (OR 3.44 with 95% CI 1.01–11.7) compared to those in the 65–74 age group (2.85 with 95% CI 1.0–6.76) and when compared to those in the 75–84 age group, injury severity score (ISS) (OR 1.08, 95% CI 1.02 to 1.15) and the presence of more than one comorbidity (OR 2.68, 95% CI 1.26 to 5.68) were independently predictive of death (at hospital discharge) on multi-variable logistic regression analysis (Table 3).

## 4. Discussion

We found that, among geriatric patients sustaining low-energy blunt trauma (falls from sitting or standing), age equal to or higher than 85, higher injury severity score (ISS) and having more than one comorbidity were predictive of death. These findings will have significant impact on the care for these fragile patients.

The older group (85+ years), in addition to having the highest mortality, had a higher percentage of patients requiring the services of acute rehabilitation hospitals and skilled nursing facilities and the smallest percentage of patients being discharged directly home, which further emphasizes the differences between this group and the two younger ones in having worse outcomes that do not only involve mortality. 

Geriatric patients are at risk of sustaining serious injuries even when the mechanism is a low-energy one. Krappinger et al. showed that these patients are at risk for arterial hemorrhage from low-energy pelvic trauma, and Shcrag et al. showed that these patients need imaging of the cervical spine when sustaining low-energy mechanism injury as clinical indicators were inadequate to rule them out since these patients are prone to serious injuries despite the mechanism [7,8]. Our data suggests that worse outcomes are present among these patients, especially among the older geriatric group. 

The older age group in this study had a statistically significant difference in race composition including a larger percentage of white patients and a smaller percentage of other races (black and “other” races) compared to the two other groups, which could have contributed to the difference in mortality in this paper as multiple authors have demonstrated differences in trauma outcomes associated with race across all age groups [9,10,11]. 

Sammy et al., in a meta-analysis of the published literature, found that while multiple factors affect the mortality of older patients, demographics (age and gender), pre-existing co-morbidities and injury severity and mechanism were significantly predictive of death [12]. A second systematic review conducted by Hashmi et al. concluded that overall mortality in geriatric trauma patients increases with age, with the 74 and older age group having twice the odds of mortality as compared to those in 65–74 age range; other studies had similar findings [13,14,15,16,17,18].Anemia, too, has been implicated as yet another risk factor for mortality [4]. Although these studies included all the mechanisms and levels of severity of injury in the older patients, our results are in agreement with their analysis as older age (75–85), higher ISS and having more than one comorbidity were associated with death in our study.

We assert that such elderly, comorbid patients should be approached with care beyond early goals-of-care conversations and standard practices. Goals of care permitting, such patients should be managed aggressively with a multidisciplinary approach to address all injuries and comorbidities.

Perhaps more importantly, fall risk assessments and preventative measures should be implemented on a community level. Numerous evidence-based assessment tools are available to identify both patient and environmental factors that can contribute to falls. Multifactorial approaches are likely required in the prevention of such traumas; such programs are themselves another area of active research.

Our study has certain limitations. First, this is a single-center, retrospective study, decreasing the generalizability of our findings. Second, the study is limited to one year; having a multi-year data would enable us to examine a larger number of patients and potentially obtain a better evaluation of factors impacting death in this patient population. Despite this, a sizable sample was still obtained. Third, there were multiple statistically significant differences between age groups in terms of injuries. While this may cause difficulties when comparing age groups, such differences are not unexpected. Pre-existing conditions in the elderly are both distributed heterogeneously and affect patient outcomes. Finally, our data set specifies low-energy blunt trauma. While this makes our results less generalizable to all geriatric trauma patients, it highlights an important subset of patients with an increasingly common mechanism injury in this age group.

## 5. Conclusions

In conclusion, geriatric trauma patients sustaining low-energy blunt trauma including falls from sitting or standing are at risk of death. Factors shown in this study to be associated with mortality on multi-variable analysis included age of 85 or more, increasing injury severity score and having more than one comorbidity. Paying close attention to geriatric trauma patients that meet these criteria could result in improved outcome. Further studies are needed to confirm these results.

## Figures and Tables

**Table 1 healthcare-10-02214-t001:** Demographic, injury, clinical characteristics and outcomes of geriatric patients with low-energy blunt trauma, N = 1285.

Demographic	
Sex, *n* (%)	
Male	447 (34.8)
Race, *n* (%)	
White	1149 (89.6)
Black	27 (2.1)
Other	107 (8.3)
Injury	
Injury severity (ISS), Median (IQR)	5 (2, 9)
Head, *n* (%)	264 (20.5)
Chest, *n* (%)	14.1 (181)
Abdomen, *n* (%)	41 (3.2)
Extremity, *n* (%)	715 (55.6)
Clinical	
ED SBP (mmHg)	148 ± 31
ED Heart rate	82 ± 17
ED Respiratory rate	19 ± 3
Hct-hematocrit	36.8 ± 6.8
INR, Median (IQR)	1.12 (1.04, 1.47)
Lactate, Median (IQR)	1.6 (1.2, 2.2)
>1 comorbidity	238 (18.5)
Smoking	63 (4.9)
AC	515 (40.1)
VAP—ventilator-associated pneumonia	2 (0.16)
PE—pulmonary embolism	1 (0.08)
DVT—deep venous thrombosis	13 (1.01)
AKI—acute kidney injury	7 (0.55)
Outcome	
LOS (days), Median (IQR)	5 (3, 7)
Disposition, *n* (%)	
Died	37 (2.9)
Hospice	10 (0.8)
SNF	243 (19)
Rehab.	337 (26.3)
Home	638 (47.8)

Rehab. = acute rehabilitation hospital, SNF = skilled nursing facility, LOS = length of stay, AC = anticoagulation, ED = emergency department, SBP = systolic blood pressure, INR = international normalization ratio.

**Table 2 healthcare-10-02214-t002:** Demographic, injury, clinical characteristics and outcomes of geriatric trauma patients with low-energy blunt trauma divided into three age groups.

	65–74 yrs. N = 254	75–84 yrs. N = 473	≥85 yrs. N = 556	*p*-Value
Demographic				
Sex, *n* (%)				
Male	95 (37.4)	184 (38.4)	168 (30.2)	0.01
Race, *n* (%)				
White	216 (85)	420 (88.8)	513 (92.3)	<0.00
Black	12 (4.7)	9 (1.9)	6 (1)	
Other	26 (10.2)	44 (9.3)	37 (6.7)	
Injury				
Severity (ISS), Median (IQR)	4.5 (2, 9)	5 (2, 9)	5 (2, 9)	0.56
Head, *n* (%)	53 (20.9)	102 (21.6)	109 (19.6)	0.73
Chest, *n* (%)	44 (17.3)	61 (12.9)	76 (13.7)	0.24
Abdomen, *n* (%)	12 (4.7)	16 (3.4)	13 (2.3)	0.19
Extremity, *n* (%)	126 (49.6)	251 (53.1)	338 (60.8)	<0.00
Clinical				
ED SBP (mmHg)	145 ± 32	146 ± 29	151 ± 32	0.56
ED Heart rate	83 ± 16	82 ± 17	81 ± 17	0.92
ED Respiratory rate	19 ± 4	19 ± 3	19 ± 4	0.01
Hct-hematocrit	36.8 ± 7.6	36.8 ± 6.3	36.7 ± 6.7	0.97
>1 morbidity	51 (20.1))	95 (20.1)	92 (16.6)	0.27
Smoking	33 (13.0)	20 (4.2)	10 (1.8)	<0.00
AC	91 (35.8)	197 (41.7)	227 (40.8)	0.28
Surgery	0	1	0	0.42
VAP—ventilator-associated pneumonia	2 (0.79)	0 (0.0)	0 (0.0)	0.02
PE—pulmonary embolism	0 (0.0)	0 (0.0)	1(0.18)	0.52
DVT—deep venous thrombosis	1 (0.39)	6 (1.27)	6 (1.08)	0.52
AKI—acute kidney injury	2 (0.79)	1 (0.21)	4 (0.72)	0.46
Outcome				
LOS (days), Median (IQR)	4 (3, 8)	5 (3, 8)	5 (3, 7)	0.37
Disposition, *n* (%)				<0.00
Died	4 (1.6)	8 (1.7)	25 (4.6)	
Hospice	2 (0.8)	2 (0.4)	6 (1.1)	
SNF	32 (12.6)	88 (18.7)	123 (22.1)	
Rehab.	59 (23.2)	123 (26.1)	155 (27.9)	
Home	149 (58.7)	244 (51.8)	244 (43.9)	
Mortality, *n* (%)	4 (1.6)	8 (1.7)	25 (4.6)	<0.01

*p*-value for ANOVA for normal distribution or Kruskal–Wallis’s test for non-normal distribution and chi-square or Fisher’s exact tests. Rehab. = acute rehabilitation hospital, SNF = skilled nursing facility, LOS = length of stay, AC = anticoagulation, SBP = systolic blood pressure, INR = international normalization ratio.

**Table 3 healthcare-10-02214-t003:** Multivariable logistic regression for factors predictive of death (at hospital discharge) among geriatric patients with low-energy blunt trauma (*n* = 1285).

	OR	95% CI	*p*-Value
Age >85 compared to			
65–74	2.78	0.94, 8.21	0.06
75–84	2.87	1.21–6.80	0.02
ISS (Injury severity score)	1.08	1.02–1.15	0.02
>1 comorbidity compared to 1 or 0	2.68	1.26–5.68	0

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
