# Peer review of "Factors Predictive of Mortality among Geriatric Patients Sustaining Low-Energy Blunt Trauma"

_healthcare, 2022, doi:10.3390/healthcare10112214_

Round 1
Reviewer 1 Report
Thank you for the opportunity to review this manuscript. Although the conclusions are not very innovative, I appreciate the efforts of the aAthors in describing low kinetic trauma in elderly patients. The sample is quite large, although the study is retrospective and monocentric. I would suggest some revisions:
- indicate in some hospital and as city and country the study was carried out
- clarify what is meant by "trauma activation"
- give a precise definition of "low-energy" trauma (for example with reference to the ATLS guidelines)
- indicate in the "methods" on what basis the patients were divided into three age groups
- clearly indicate the outcomes in the "methods" section (eg survival at 30 days and / or at hospital discharge)
- being clear about the time at which mortality was assessed is very important; is mortality believed to be due to the trauma itself? or the medical consequences (eg infections) of trauma and hospitalization? this aspect should be discussed extensively
- it would be advisable to give information on imaging, eg how many patients have undergone CT?
- if possible, I would better define the type of injury: which cranial injuries? which thoracic? which abdominal?
- I believe it is useful to report the age data also continuously, and not only categorized into the three groups; it could be useful to report a multivariate model with age considered as a continuous variable
- in the tables report the unit of measurement (eg SBP: mmHg ...)
Author Response
Thank you for taking the time to review our manuscript, the comments have been itemized with the responses listed below.
- Indicate in some hospital and as city and country the study was carried out
- We thank the reviewer for the comment, it has been revised.
- After obtaining the Institutional Review Board (IRB) approval, a retrospective analysis of geriatric trauma patients (65 years and older) who arrived to our trauma center (Staten Island University Hospital Center in Staten Island, New York, USA) as trauma activation between January 2019 and October 2019 was conducted using the trauma database.
- Clarify what is meant by "trauma activation"
- We thank the reviewer for the comment, it has been revised.
- Hospital alert for highest level of critical care: mobilizing entire surgical trauma service, Emergency department, nursing, and ancillary staff
- Give a precise definition of "low-energy" trauma (for example with reference to the ATLS guidelines)
- We thank the reviewer for the comment, it has been revised.
- Low energy trauma involves falls from a ground-level height, including standing, sitting
- Indicate in the "methods" on what basis the patients were divided into three age groups
- We thank the reviewer for the comment, it has been revised.
- Age is an independent predictor of death in trauma patients, we wanted to assess the impact age has on outcomes of geriatric (65 years of age and older) patients by categorizing them into three age groups for ease of interpretation and assessment into 65-74, 75-84 and 85+. Having the patients belong to three 10-year categories enables ease of interpretation and management.
- Clearly indicate the outcomes in the "methods" section (e.g. survival at 30 days and / or at hospital discharge
- We thank the reviewer for the comment, it has been revised to reflect survival at hospital discharge.
- This is a retrospective cohort study. Categorical data were summarized by the number and percentage of patients falling within each category. Continuous variables were summarized by descriptive statistics including mean and standard deviation or median and interquartile range. The primary comparison group is age group (65-74, 75-84, 85+). Age is an independent predictor of death in trauma patients, we wanted to assess the impact age has on outcomes of geriatric (65 years of age and older) patients by categorizing them into three age groups for ease of interpretation and assessment into 65-74, 75-84 and 85+. Having the patients belong to three 10-year categories enables ease of interpretation and management. The primary outcome variable is hospital mortality. Bivariate analyses were performed using the χ2-test, ANOVA and Wilcoxon Rank-sum test, as appropriate. The independent effects of age group, Injury Severity Score (ISS) and previous medical history (whether patient has two or more comorbidities) were evaluated using a multivariable logistic regression analysis. All statistical tests were two-sided. P-values <0.05 were considered statistically significant. All statistical analyses were performed using SAS software (Statistical Analysis Systems Inc., Cary, NC, USA). Confidence intervals (CIs) will be two-sided, unless otherwise stated. This was done to evaluate if age and the presence of more than one co-morbidity were independently predictive of death (at hospital discharge) on multi-variable logistic regression analysis
- Being clear about the time at which mortality was assessed is very important; is mortality believed to be due to the trauma itself? or the medical consequences (eg infections) of trauma and hospitalization? This aspect should be discussed extensively
- We thank the reviewer for the comment, it has been revised.
- 1285 geriatric patients (age >65 years) presented to our level 1 trauma center as trauma activation (hospital alert to mobilize surgical trauma service, emergency department trauma team, nursing, and ancillary staff for highest level of critical care) with a low-energy blunt trauma (fall from sitting or standing) during the study period. 34.8% of the patients were Men, 89.6% were White and 20.5% had at least one comorbidity. The median injury severity score (ISS) was 5 and the median hospital length of stay (LOS) was 5 too. Injury and clinical characteristics are demonstrated in Table 1, thirty-seven (2.9%) patients died with mortality determined to be due to the trauma itself.
- It would be advisable to give information on imaging, e.g. how many patients have undergone CT?
- We thank the reviewer for the comment, it has been revised.
- After obtaining the Institutional Review Board (IRB) approval, a retrospective analysis of geriatric trauma patients (65 years and older) who arrived to our trauma center (Staten Island University Hospital Center in Staten Island, New York, USA) as trauma activation (hospital alert to mobilize surgical trauma service, emergency department trauma team, nursing, and ancillary staff for highest level of critical care) between January 2019 and October 2019 was conducted using the trauma database. All patients received computed tomography scans (CT Scans) of the head, cervical-spine, chest, abdomen and pelvis.
- If possible, I would better define the type of injury: which cranial injuries? which thoracic? which abdominal?
- We thank the reviewer for the comment, but for the purposes of this study did not define the specific cranial/thoracic/abdominal injuries.
- I believe it is useful to report the age data also continuously, and not only categorized into the three groups; it could be useful to report a multivariate model with age considered as a continuous variable
- We thank the reviewer for the comment, we believe that the categorization of the patients into three age groups clearly shows the impact each decade of age on the survival of these patients. Age as a continuous variable has been extensively studied in the literature in trauma patients.
- In the tables report the unit of measurement (eg SBP: mmHg ...)
- We thank the reviewer for the comment, it has been revised

Reviewer 2 Report
the paper is interesting and its scope is coherent to the scope of the journal.
Authors should better classify the patients included in the study using international classifications (Ingravallo F, Cerquetti I, Vignatelli L, Albertini S, Bolcato M, Camerlingo M, Corbi G, De Leo D, De Nicolò A, De Stefano F, Dell'Erba A, Di Giulio P, Domenici R, Fedeli P, Feola A, Ferrara N, Forti P, Frigiolini F, Gianniti P, Gili E, Iannone P, Lovato A, Lunardelli ML, Marengoni A, Marozzi F, Martelloni M, Mecocci P, Molinelli A, Polo L, Portas M, Rossi P, Scorretti C, Trabucchi M, Volpato S, Zoja R, Castellani GL. Medico-legal assessment of personal damage in older people: report from a multidisciplinary consensus conference. Int J Legal Med. 2020 Nov;134(6):2319-2334. doi: 10.1007/s00414-020-02368-z. Epub 2020 Jul 17. PMID: 32681208; PMCID: PMC7578136).
It could be also of interest understand if after the blunt trauma there is a loss of skills of the patient using ADL or IADL scale.
In the methodology part they should better classify the trauma (low energy and high energy) this concept should be better analyzed.
References should be formatted according to the guidelines of the journal.
Author Response
We thank you for taking the time to review our manuscript. The comments have been itemized and addressed as below.
- Authors should better classify the patients included in the study using international classifications (Ingravallo F, Cerquetti I, Vignatelli L, Albertini S, Bolcato M, Camerlingo M, Corbi G, De Leo D, De Nicolò A, De Stefano F, Dell'Erba A, Di Giulio P, Domenici R, Fedeli P, Feola A, Ferrara N, Forti P, Frigiolini F, Gianniti P, Gili E, Iannone P, Lovato A, Lunardelli ML, Marengoni A, Marozzi F, Martelloni M, Mecocci P, Molinelli A, Polo L, Portas M, Rossi P, Scorretti C, Trabucchi M, Volpato S, Zoja R, Castellani GL. Medico-legal assessment of personal damage in older people: report from a multidisciplinary consensus conference. Int J Legal Med. 2020 Nov;134(6):2319-2334. doi: 10.1007/s00414-020-02368-z. Epub 2020 Jul 17. PMID: 32681208; PMCID: PMC7578136).
- We thank the reviewer for the comment. This is a retrospective cohort study. Categorical data were summarized by the number and percentage of patients falling within each category. Continuous variables were summarized by descriptive statistics including mean and standard deviation or median and interquartile range. The primary comparison group is age group (65-74, 75-84, 85+). Age is an independent predictor of death in trauma patients, we wanted to assess the impact age has on outcomes of geriatric (65 years of age and older) patients by categorizing them into three age groups for ease of interpretation and assessment into 65-74, 75-84 and 85+. Having the patients belong to three 10-year categories enables ease of interpretation and management.
- It could be also of interest understand if after the blunt trauma there is a loss of skills of the patient using ADL or IADL scale.
- We thank the reviewer for the comment, it is not possible for us to address this in this study.
- In the methodology part they should better classify the trauma (low energy and high energy) this concept should be better analyzed.
- We thank the reviewer for the comment, it has been revised.
- Low energy trauma involves falls from a ground-level height, including standing, sitting
- References should be formatted according to the guidelines of the journal.
- We thank the reviewer for the comment, it has been revised
